# The Improvement of Kaolinite Supported Cerium Oxide for Styrene–Butadiene Rubber Composite: Mechanical, Ageing Properties and Mechanism

**DOI:** 10.3390/polym14235187

**Published:** 2022-11-29

**Authors:** Honglei Liu, Kaiyuan Xiao, Yinmin Zhang, Yanbing Gong, Yongfeng Zhang

**Affiliations:** 1Chemical Engineering College, Inner Mongolia University of Technology, Hohhot 010051, China; 2Inner Mongolia Key Laboratory of Efficient Recycle Utilization for Coal-Based Waste, Inner Mongolia University of Technology, Hohhot 010051, China

**Keywords:** kaolinite, cerium oxide, styrene–butadiene rubber, mechanical property, ageing property

## Abstract

Kaolinite supported cerium oxide (CeOx/Kaol) was successfully prepared via a deposition method and used to improve the mechanical and aging properties of styrene–butadiene rubber (SBR) composite. The scanning electron microscopy (SEM) and transmission electron microscopy (TEM) results showed that cerium oxide has a successfully loading and fine distribution on the edge and surface of kaolinite. Fourier transform infrared (FT-IR) spectroscopy indicated that cerium oxide may interact with the surface hydroxyls of kaolinite. The CeOx/Kaol material had a uniform dispersion in the resulting SBR composite. The loading of cerium oxide on Kaol increases the scorch time (*t*_10_) and curing time (*t*_90_) of the filled SBR composites relative to the pure SBR. The mechanical parameters of the filled SBR composites were increased significantly. The tensible strength and tear strength at 40 phr content with 4% CeOx loading reached 12.85 Mpa and 51.16 kN/m, which were increases of 35.9% and 38.3%, respectively, relative to that of the SBR filled with raw Kaol. The anti-ageing characteristic of the resulting composite showed an obvious improvement with the loading of CeOx. Meanwhile, the reinforcement and anti-ageing mechanisms of the CeOx/Kaol were proposed. These results were attributed to the complexation between Ce elements on the surface of Kaol and rubber chains through a double bond. This could improve the incorporation between rubber molecules and filler particles, and restrict rubber chain motion via trapping rubber chains.

## 1. Introduction

Rubber, as a particular class of polymer materials, is widely used in industrial and civilian fields because of its unique characteristics [1,2,3,4]. Nowadays, styrene–butadiene rubber (SBR), as a typical type of rubber, has particularly been recognized for its good abrasion performance and water resistance [5]. Therefore, SBR is still of great technical importance for the tire rubber industry. However, SBR has some disadvantages, especially in terms of its poor mechanical and anti-ageing properties. Static mechanical and anti-ageing properties of rubber products are considered to be important properties for many engineering applications, including tire products, cable materials, vibration absorbers, etc., [6,7]. In recent years, styrene–butadiene rubber (SBR) composites have received important research attention, because their material properties have been significantly improved compared to the original polymer. Adding filler to SBR can significantly improve the mechanical properties, thermal properties, barrier properties, and dynamic mechanical properties of SBR composites. Many studies on the static mechanical and anti-ageing properties of rubber vulcanisates have been reported, including on the effects of type fillers and the interaction between rubber and filler [8,9,10,11].

Clay minerals are very effective in polymer structures, because they are small in particle size, intercalated, and hydrated layered aluminosilicates with reactive -OH groups on the surface [12]. Clay mineral/rubber composites have recently attracted significant research interest because they present significantly improved actual properties when compared with virgin polymers [13,14]. Kaolinite is a typical 1:1 (two sheets) type clay mineral, consisting of a tetrahedral sheet of SiO_2_ siloxane units and an octahedral sheet of AlO_2_ (OH)_4_ [15,16,17,18]. Kaolinite is suitable for use as functional filler for rubber composites because of its whiteness and fine particle size. In addition, it also has good plasticity, good acid solubility resistance, low cation exchange capacity, high dispersion, and other physical and chemical properties. It is suitable for use as a functional filler in rubber composites, which can give rubber excellent properties, especially in elasticity, flexure resistance, better dimensional stability, barrier performance, elongation at break, compression deformation, and other aspects [1,19]. Kaolinite modified with appropriate agents can have compatibility and dispersibility within a rubber matrix. Good dispersion of the clay layer in the polymer matrix is very important to obtain qualified clay-based polymer nanocomposites [20], and can significantly improve properties of the filled rubber composite, such as its mechanical, thermal, barrier and anti-ageing properties [21].

Rare earth (RE) was widely used in the preparation of functional materials because of its special electronic structure [22]. The polymer composites can be modified through rare earth because rare earth ions possess many unoccupied orbits, have strong oxidation-resistant ability, and have great scavenging effects for free radicals [23,24]. The characteristics of rare earth could retard the autoxidation processes effectively and improve crosslinking intensity, ageing characteristics, accelerating vulcanization reactions, etc., [25]. However, it is important to realize uniform dispersion and prevent the aggregation of rare earth particles in polymer matrix. Many researchers have reported that the rare earth ions or oxidation loaded on the inorganic particles have modified polymer composites, including carbon, CaCO_3_, SiO_2_, etc., [10,26,27,28]. However, the reinforcement of rare earth, especially cerium-oxide-modified kaolinite, and the reinforcement of SBR composites have rarely been reported, and improving the dispersion of cerium oxide, avoiding its agglomeration in the rubber matrix, and achieving a slow and continuous release of rare earth properties are still difficult points to study.

This work proposed a new approach to modify the surface characteristic of kaolinite through cerium oxide via chemical deposit method. The CeOx particles were successfully loaded on the surface and edge of the kaolinite and had a uniform dispersion. The effects of CeOx content and functional filler content on the vulcanization, mechanical properties, and ageing stability of SBR composite were investigated. This work provided a new design approach for the functional utilization of clay minerals and the improvement of rubber composites.

## 2. Experimental

### 2.1. Material

The kaolinite powder in this study was obtained from Inner Mongolia, China. The Kaol sample was wet grounded, Kaol sample particle size distributions were analyzed with a BT9600S laser particle size analyzer (Bettersize Instruments Co., Ltd., Liaoning, China.) and the *d_50_* was determined as 1.1μm, which is named as Kaol. The Ce (NO_3_)_3_·6H_2_O was purchased from Shanghai Macleans Biochemical Technology Co., Ltd., shanghai, China. The CaCl_2_ was purchased from Tianjin Fengchuan Chemical Reagent Technology Co. Ltd., Tianjin, China. The solid styrene–butadiene rubber (SBR) was purchased from Gaoshike industrial and trade Co., Ltd., Shandong, China. The accelerator N-tert-buylbenzothiazole-2-sulphenamide (NS), zinc oxide (ZnO), stearic acid (SA), and sulphur were from Sino-pharm Chemical Reagent Co., Ltd., Beijing, China. The deionized water was prepared in laboratory.

### 2.2. Preparation of CeOx/Kaol Material and CeOx/Kaol/SBR Composites

Here, 5 g Kaol samples were added into 50 mL deionized water to prepare a suspension with a mass fraction of 20%. The suspension system was stirred for 30 min. A Ce (NO_3_)_3_·6H_2_O solution with 0.1 mol/L and 2 mL H_2_O_2_ was added into the suspension system. The loading content of CeOx was controlled via changing the mass ratio of Ce/Kaol. The pH value of the mixture system was adjusted to 9 by 1 mol/L ammonia water at the rate of 10 mL/min. Then the mixture system was stirred at ambient temperature for 40 min in the condition of ultrasonic bath. The obtained mixture solution was centrifugal filtered, rinsed with absolute ethyl alcohol three times, and dried in a vacuum at 105 °C for 12 h. Finally, the product was calcined in a muffle oven at 500 °C in air for 2 h (heating rate 5 °C/min).

The CeOx/Kaol/SBR composite was prepared via latex blending as follows: the CeOx/Kaol material was dispersed in deionized water and stirred for 30 min to prepare uniform suspension. The prepared suspension was mixed with SBR latex via a mechanical mixer. The 10% CaCl_2_ aqueous solution was added into the mixture system using a peristaltic pump with a rate of 10 mL/min. During the stirring process, the pH was kept constant by slowly dropping NH_3_·H_2_O and using a pH meter to measure the pH value of the system. The pH value of this mixture system was maintained at 11.0. Finally, the obtained mixture was dried at 70 °C until constant mass. The dried sample is thinly plasticized in an open mill at room temperature using a water cooling system. The spacing between the two rolls was about 0.15 cm, and the roll rate was 6.98 m/min. Then, ZnO, stearic acid (SA), accelerator (NS), and sulfur were successively added into the block, composition formula is shown in Table 1. The Ce/Kaol/SBR composite sample was vulcanized in a 25TQLB vulcanizer at 160 °C and 103MPa at the best curing time.

### 2.3. Characterization

The X-ray diffraction (XRD) patterns were obtained by using a SmartLab rotary cathode X-ray diffractometer with Cu Kα radiation (λ = 1.540596 nm) generated at 40 kV and 100 mA. The sample was scanned from 5° to 70° with a speed of 10° min^−1^.

Fourier infrared spectra (FT-IR) of samples were recorded using a Fourier transform infrared spectrometer (Magna-IR 750 Nicolet, Madison, WI, USA) at a resolution of 4 cm^−1^ from 4000 to 400 cm^−1^. The number of scans accumulated was 32.

The SEM images of the composite sample were obtained via a Hitachi SU8020 cold field emission scanning electron microscope. The prepared rubber composites were characterized using a JEM-2100 transmission electron microscope with an acceleration voltage of 200 kV. Samples were prepared by ultramicrotomy of the bulk cured composites to give a section of about 50 nm thickness.

The transmission electron micrograph (TEM, FEI Tecnai G2 F20) that was used to characterize the surface morphology of the resulting samples operated at an accelerating voltage of 80 kV.

## 3. Results and Discussion

### 3.1. The Characterization of the Prepared CeOx/Kaol Materials

The crystal structure change of Kaol loading was evaluated by XRD. The XRD patterns of Kaol, CeOx, and CeOx/Kaol samples are shown in Figure 1. Two obvious diffraction peaks at 2θ = 12.3° and 25.4° with the values of 0.714 nm and 0.357 nm are attributed to the (001) and (002) lattice plane of Kaol [18,29,30], respectively. There are three diffraction peaks at 2θ = 18° to 24° with the values of 0.446, 0.434 and 0.416nm, which are attributed to the (020), (110), and (111) lattice plane, respectively. Several diffraction peaks at 2θ = 35° to 40° for Kaol presented favorable separation condition and peak shape. Meanwhile, two weak diffraction peaks at 2θ = 26.7° and 62.1° indicated the presence of quartz [31]. Three obvious diffraction peaks for CeOx were observed at 2θ = 28.2°, 47.8°, 56.3° in the XRD of CeOx/Kaol, which indicated the presence of CeOx [32]. However, the characteristic diffraction peaks attributed to (001) and (002) lattice plane of Kaol disappeared after the loading of CeOx, and two diffraction peaks approximately at 2θ = 20° were observed. The results indicated that the loading of CeOx could affect the crystal structure of Kaol, and the increase in the adhesion area of CeOx particles is conducive to the uniform dispersion of CeOx. In addition, after grinding and stripping, the lamellar structure of kaolin changes strongly along the c-axis, leading to the disorder of the crystal structure of kaolinite, particle size decreases, and the specific surface area increases [33,34,35].

The FT-IR spectra of Kaol and CeOx/Kaol material are displayed in Figure 2. In the high wavenumber region, there are two obvious bands at 3695.35 cm^−1^ and 3619.77 cm^−1^; these were attributed to the stretching vibration of inner surface hydroxyl and inner hydroxyl, respectively [36,37]. In the medium wavenumber region, the three bands at 1114.71 cm^−1^, 1031.84 cm^−1^, and 1008.07 cm^−1^ presented favorable separation, which are associated with the stretching vibration of Si-O in the silica tetrahedron, which indicated that the Kaol sample presented a high crystallinity and had an agreement with the XRD data [38,39,40]. In the low wavenumber region, the band at 914.7 cm^−1^ is attributed to the bending vibration mode of Al-OH. The bands observed at 541.4 cm^−1^ and 472.5 cm^−1^ are attributed to the bending vibration of Al-O-Si. With the loading of CeOx, the intensities of characteristic bands for hydroxyl groups decreased significantly in the high wavenumber. Meanwhile, in the middle wavenumber, a wide band was observed approximately at 1000 cm^−1^, which is attributed to the stretching vibration of Si-O and indicated that the crystallinity of Kaol decreased due to loading of CeOx. In the low wavenumber, the characteristic bands for the bending vibration mode of Al-OH and Al-O-Si all changed weakly and widely, which also indicated the decrease of Kaol, and is in agreement with the above data.

Figure 3 displays the XPS full spectrum and Ce 3d spectrum of CeOx/Kaol material. As shown in the Figure 3a, the characteristic peaks for Ce 3d were observed in the XPS full spectrum CeOx/Kaol material, which indicated that the CeOx particles were successfully loaded on the surface of the Kaol. As shown in the Figure 3b, the Ce 3d spectrum binding energy were fitted by seven peaks approximately at 882.64 eV, 886.02 eV, 889.04 eV, 898.26 eV, 901.31 eV, 906.93 eV, and 916.70 eV [26]. The binding energy peaks at 882.64 eV, 889.04 eV, 898.26 eV, 901.31 eV, 906.93 eV, and 916.70 eV were assigned to the Ce^IV^, and the binding energy peak at 886.02 eV was attributed to the Ce^III^ [21,41]. The contents of Ce^IV^ and Ce^III^ were calculated and determined as 88.63% and 11.37%, which indicated that Ce^IV^ was the most abundant cation in the bulk of CeOx loading on the Kaol.

The SEM images and EDS of Kaol and Ce/Kaol material are shown in Figure 4. The Kaol presented a stacked lamellar microstructure with a thickness of 1 µm ~ 2 µm. The edge of the lamellar structure showed as relatively flat. Many lamellar layers of Kaol were dispersed on the edge of the aggregates. When the Kaol was loaded by CeOx, obvious small particles were observed on the transverse edge of Kaol lamellar structure. The surface and edge of Kaol gradually changed to be rough. The EDS and electro images indicated that the CeOx presented a dispersive and uniform distribution on the surface and edge of the Kaol. The TEM images of the Ce/Kaol material are shown in Figure 5. The CeOx particles were clearly observed on the surface and edge of the Kaol lamellar structure with the particle size of 5 ~ 10 nm. As shown in Appendix A, there was no obvious aggregated phenomenon observed in the whole view. Meanwhile, the lattice fringe of the CeOx particle on the edge of Kaol lamellar structure was characterized and analyzed. The (111) and (200) lattice plane were observed with the characteristic values of 0.317 nm and 0.270 nm.

### 3.2. Characterization of the Prepared CeOx/Kaol Filled SBR Composites

#### 3.2.1. The Microstructure of the Prepared CeOx/Kaol Filled SBR Composites

The SEM images of SBR composites filled with CeOx/Kaol material are shown in Figure 6. The CeOx/Kaol particles had a uniform and fine dispersion in the SBR matrix. There was no significant aggregation in the overall view. Meanwhile, the rubber chains interacted with CeOx/Kaol particles, and which were confined within the interparticle space of CeOx/Kaol particles. This could restrict the motion of rubber molecule chains and prevent the extension of the cracks when the resulting composite is stretched. Figure 7 shows the TEM images of SBR composites filled with CeOx/Kaol material. The CeOx/Kaol material with the layer-like particles exhibited significant characteristics with a diameter from 300 nm to 500 nm and a thickness from 50 nm to 200 nm. The average distances between the layer-like particles ranged from dozens of nanometers to hundreds of nanometers. As shown in Appendix A, the CeOx particles with a dozen nanometers were observed on the surface of Kaol and interacted with the rubber chains. Similar to previous works, the phenomenon of bent lamellar structures was observed, which may be due to mechanical action during processing of rubber composites.

#### 3.2.2. The Processing Properties of the Prepared CeOx/Kaol Filled SBR Composites

The processing characteristics of SBR composites filled with CeOx/Kaol material loaded with different contents of CeOx are shown in Table 2. The cure parameters including the scorch time (*t*_10_), the curing time (*t*_90_), the maximum torque (*M_H_*), and the minimum torque (*M_L_*) were used to evaluate the processing characteristic of the resulting composites [2,42]. The *M_H_ and M_L_* of the resulting composites had an obvious increase versus the pure SBR composite. Meanwhile, the *M_H,_* and *M_L_* exhibited an increase trend when the CeOx contents increased from 0 to 4%. The *ΔM* reached a maximum value of 5.9 dN·m when the CeOx content was 4%. The *t*_10_ and *t*_90_ for SBR composites filled with CeOx/Kaol complex had an obvious increase versus the pure SBR composite, which reached a maximum value of 5.4 min and 21.4 min, respectively, when the CeOx content was 3%. Table 3 shows the cure parameters of SBR filled with different contents of CeOx/Kaol material (CeOx loading content is 4%). The *M_H_* and *M_L_* of the resulting composites increased significantly with increasing CeOx/Kaol material content; these values reached 13.3 dN·m and 2.3 dN·m, respectively, when the filler content was 40 phr. The *t*_10_ and *t*_90_ gradually increased as the content (phr) of CeOx/Kaol material increased, which may be attributed to the interaction between CeOx/Kaol and auxiliaries, which then affected the curing process. These data are attributed to the incorporation of rubber chains between the CeOx/Kaol particles, and the CeOx particles on the surface of Kaol can improve the physical interactions of filler particles and rubber molecule chains.

#### 3.2.3. The Mechanical Properties of the Prepared CeOx/Kaol Filled SBR Composites

The static mechanical properties of pure SBR and CeOx/Kaol material filled SBR composites are shown in Table 4 and Table 5. Tear strength increased significantly as the CeOx loading content increased from 0 to 4%; this result is in agreement with the data of *MH*, *ML*, and *ΔM.* However, the tear strength showed a decrease trend when the CeOx loading content exceed 4%. The maximum tear strength value of 51.16 kN/m was an increase of 410.6% and 38.3% compared to pure SBR and SBR composite filled with raw Kaol, respectively. The tensible strength, 100% tensile modulus, 300% tensile modulus, and 500% tensile modulus exhibited similar change tendencies. The tensible strength reached the maximum value of 12.85 Mpa, which was an increase of 811.3% and 35.9% compared to pure SBR and SBR composite filled with raw Kaol, respectively. The hardness had no obvious change with increasing CeOx loading content. These data indicate that the reasonable loading of CeOx on the surface of Kaol could improve the interaction between filler particles and rubber chains, thus restricting the motion of rubber chains and improving the mechanical parameters of filled SBR composites. Table 5 shows the static mechanical properties of SBR filled with different contents of CeOx/Kaol material. There is an increase in hardness with increasing CeOx/Kaol contents. The tensible strength and tear strength had a significant increase when the CeOx/Kaol content increased from 10 phr to 50 phr. These results could be attributed to the fine dispersion of CeOx/Kaol particles in the rubber matrix and the strong interactions between filler particles and rubber chains, which constrained the motion of the rubber chains when the resulting composites were stretched. However, the mechanical characteristic overall presented gradually decreasing tendencies as the filler content changed from 50 phr to 80 phr, which may be attributed to the aggregation of CeOx/Kaol particles in the rubber matrix system at high content. These data indicated that CeOx/Kaol material can significantly reinforce the mechanical properties of the filled SBR composites at reasonable CeOx loading content and filler content. First, the loading of CeOx particles changes the surface property of Kaol. The interaction between rare earth elements and the double bonds in the rubber chains due to 4*f* shell structure could improve the incorporation between rubber molecules and filler particles, which could improve the restriction on the motion of rubber molecules during the stretching of filler composites. Second, the inorganic Kaol particles with rigid properties having strong interactions in the rubber matrix could prevent the propagation of cracks during the tearing process. Third, many rubber chains trapped between the layer-like particles forming the bonded rubber possessed a rigidity similar to the kaolinite filler. Therefore, increased filler content in the composite could carry much more stress and improve the tensile strength. The reinforcement mechanism is displayed in Figure 8.

#### 3.2.4. The Ageing Behaviors of the Resulting SBR Composites

The ageing behaviors and relative ultimate mechanical parameters of the pure SBR and the resulting SBR composites are shown in Table 6. The tensible strengths all gradually decreased as the exposure time increased from 0 h to 72 h for the pure SBR, the Kaol/SBR composite, and the CeOx/Kaol/SBR composite. However, the tendency towards decrease in the three samples presented differently. Due to the lack of filler, the pure SBR had a lower initial value of tensible strength and decrease rate relative to the Kaol/SBR composite and CeOx/Kaol/SBR composite. The Kaol/SBR composite had the most conspicuous decrease in the tensible strength, which decreased from 9.45 Mpa to 3.79 Mpa, or a 59.89% decrease. With the loading of CeOx on the surface of Kaol, the decrease rate of tensible strength decreased with increasing exposure time relative to the Kaol/SBR composite, which indicated that the loading of CeOx could improve the ageing properties of filled SBR composite. The microstructure for the filled SBR composite is shown in Figure 9. Many holes were observed in the overall view, which resulted in the reduction of the crossing link and then a decrease in the mechanical parameters. Meanwhile, an increase in the Shore (A) hardness value of SBR versus the ageing time exposure was observed, and the SBR composite filled with CeOx/Kaol had the most obvious increase tendency. These results are in accordance with those found in the literature [43,44]. This result was attributed to the reduction in the mobility of the chains [45,46]. Combined with the data of microstructure and mechanical characteristics of the resulting composites, the ageing mechanism of CeOx/Kaol for SBR composite could be demonstrated as follows: the interface interaction between the rubber matrix and the filler particles plays a vital role for the mechanical properties of the obtained composites, which is affected by the surface characteristics of the filler particles. With the functionality of the Kaol lamellar surface, the rubber molecule chains could interact with the functional groups loaded on the surface of the Kaol particles. The Ce element could complicate with the SBR chains through the double bonds due to 4*f* shell structure [47], which could improve the incorporation between rubber molecules and Kaol lamellar particles; this can restrict the motion of the rubber molecules and the action between the rubber molecules and the oxygen molecules during ageing process, thus improving the anti-ageing properties.

## 4. Conclusions

The CeOx/Kaol material was successfully prepared via a deposition method and used to improve the mechanical and aging properties of SBR composite. The effects of cerium oxide loading content and filler content on mechanical and ageing properties of the resulting SBR composites were evaluated. The SEM and TEM data showed that cerium oxide has a successfully loading and fine dispersion on the edge and surface of the Kaol, and the CeOx/Kaol material had a uniform dispersion in the resulting SBR composite. The loading of cerium oxide increases the scorch time (*t*_10_) and curing time (*t*_90_) of the filled SBR composites relative to the pure SBR. The mechanical parameters of the filled SBR composites were increased significantly. The tensible strength and tear strength at 40 phr content with 4% CeOx loading reached 12.85 Mpa and 51.16 kN/m, which was an increase of 35.9% and 38.3% relative to that of the SBR filled with raw Kaol, respectively. The anti-ageing characteristic of the resulting composite had an obvious improvement with the loading of CeOx. These results were attributed to the complexation between Ce elements on surface of Kaol and rubber chains through double bond. This could improve the incorporation between rubber molecules and filler particles, and restrict the rubber chain motion via trapping rubber chains. In this paper, the preparation, vulcanization properties, mechanical properties and aging stability of the composite are mainly studied. In the later stage, the gas barrier and thermal stability of the composite can be further studied.

## Figures and Tables

**Figure 1 polymers-14-05187-f001:**
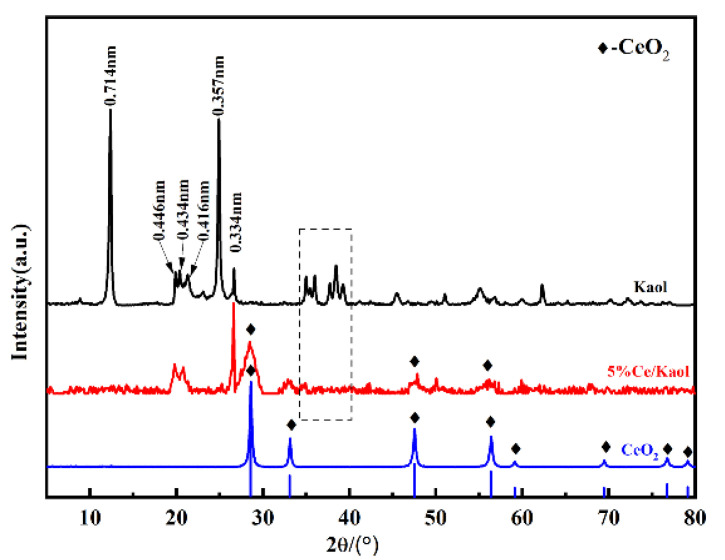
XRD patterns of Kaol and CeOx/Kaol.

**Figure 2 polymers-14-05187-f002:**
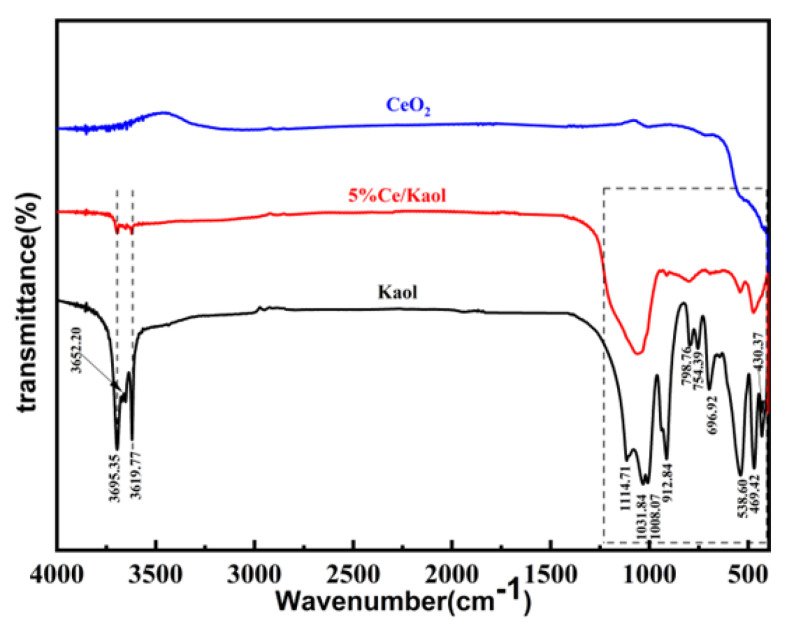
FT-IR spectra of Kaol and CeOx/Kaol.

**Figure 3 polymers-14-05187-f003:**
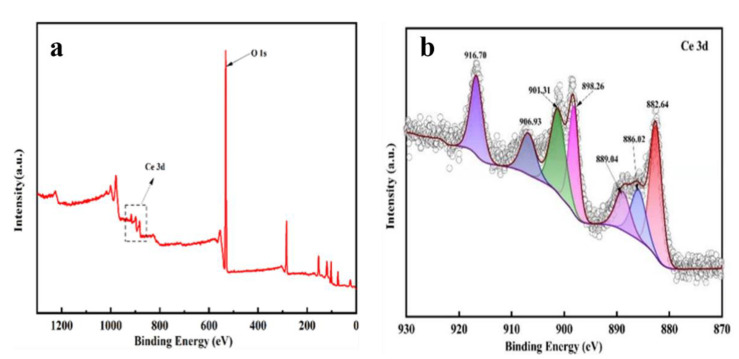
XPS analysis of (**a**) Survey spectrum of CeOx/Kaol material and high resolution spectrum of (**b**) Ce 3d.

**Figure 4 polymers-14-05187-f004:**
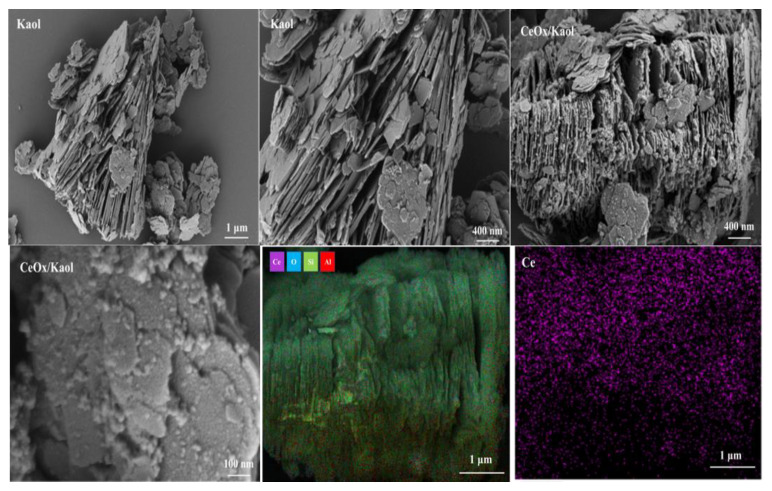
SEM images and EDS of Kaol and CeOx/Kaol material.

**Figure 5 polymers-14-05187-f005:**
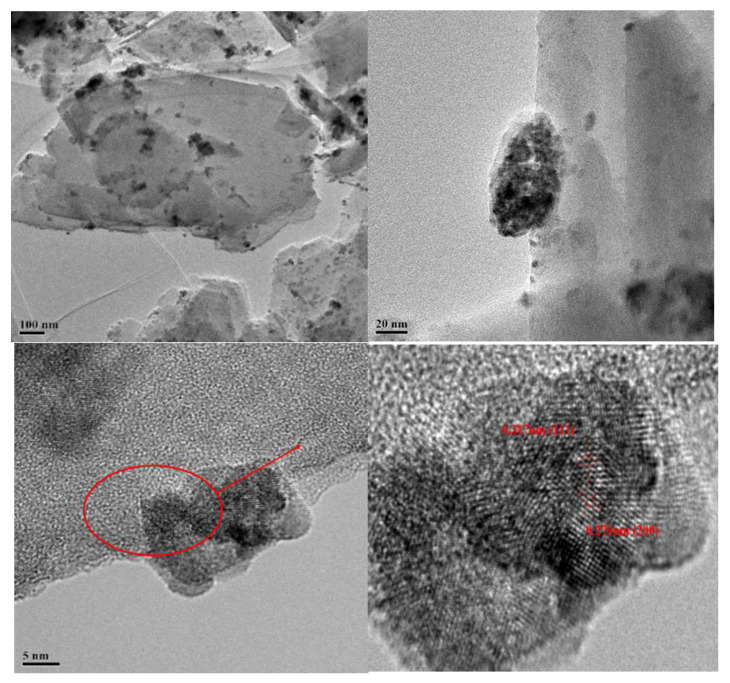
TEM image of CeOx/Kaol material.

**Figure 6 polymers-14-05187-f006:**
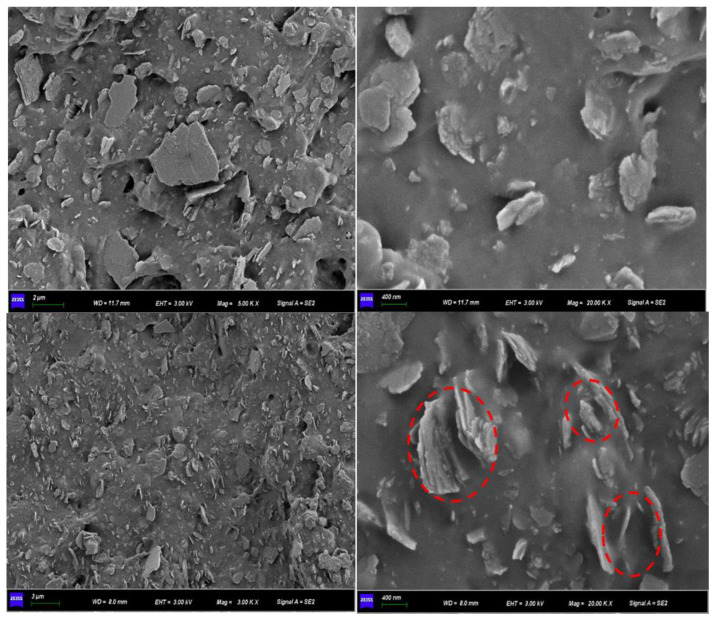
SEM image of SBR composite filled with CeOx/Kaol material.

**Figure 7 polymers-14-05187-f007:**
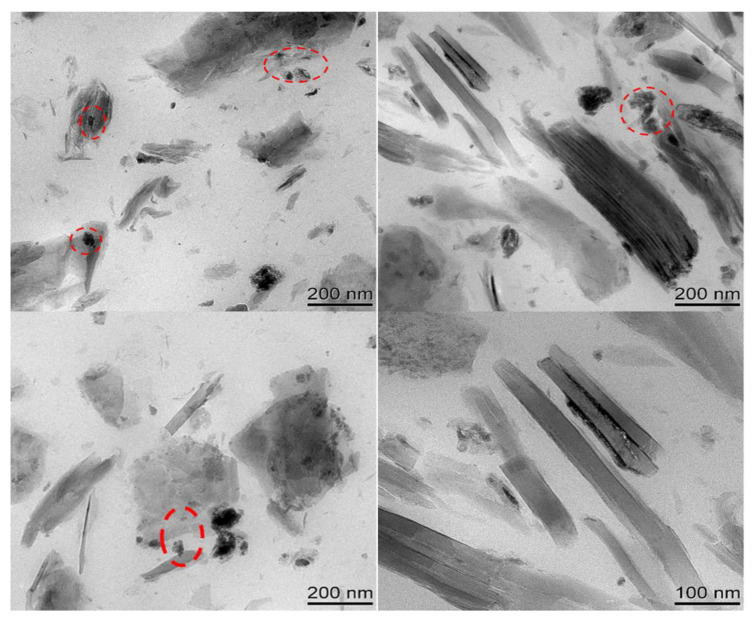
TEM image of SBR composite filled with CeOx/Kaol material.

**Figure 8 polymers-14-05187-f008:**
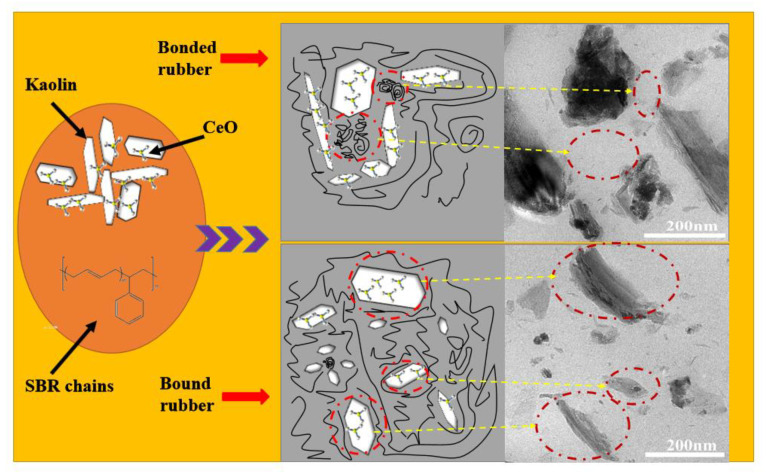
Reinforcement mechanism of the prepared composites.

**Figure 9 polymers-14-05187-f009:**
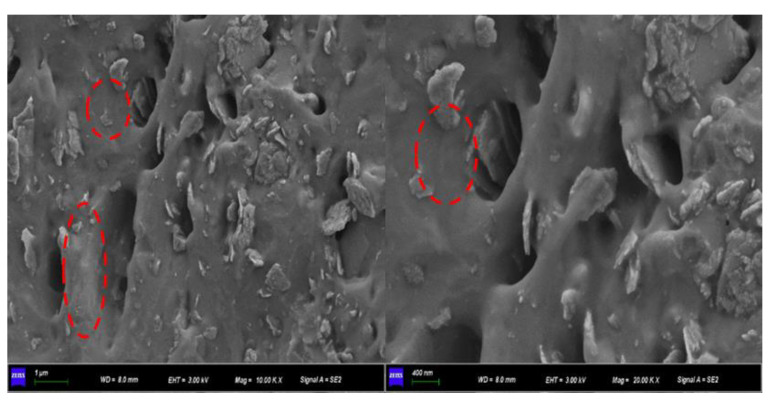
SEM image of CeOx/Kaol material filled SBR composite after aging 72 h.

**Table 1 polymers-14-05187-t001:** Composition formula of CeOx /Kaol /SBR composites.

Ingredient	E-SBR	Ce/Kaol	ZnO	SA	NS	Sulfur
Content/phr	100.00	Variable	3.00	1.00	1.00	1.75

**Table 2 polymers-14-05187-t002:** Processing properties of SBR composites filled with CeOx/Kaol complex loaded different contents of CeOx.

CeOx (%)	*M_H_* /(dN·m)	*M_L_* /(dN·m)	*ΔM* /(dN·m)	*t_10_* /(min)	*t_90_* /(min)
Pure SBR	3.30	0.40	2.90	3.4 ± 0.6	14.5 ± 0.7
0	6.70	0.90	5.80	3.5 ± 0.4	17.3 ± 0.9
1	6.40	1.00	5.40	5.0 ± 0.7	19.2 ± 0.6
2	6.70	1.10	5.60	5.4 ± 0.6	21.4 ± 0.9
3	6.40	1.10	5.30	5.3 ± 0.9	21.3 ± 0.6
4	7.10	1.20	5.90	5.4 ± 0.9	19.0 ± 0.7
5	5.90	1.00	4.90	6.0 ± 0.5	19.4 ± 0.3
6	6.00	1.30	4.70	6.0 ± 0.7	19.1 ± 0.8
7	5.90	1.20	4.70	6.4 ± 0.6	19.1 ± 0.9
8	6.00	1.40	4.60	6.3 ± 0.7	20.3 ± 0.3

*ΔM*= *M_H_ − M_L_*; Pure SBR: no filler is added; 0 CeOx: Raw Kaol; 1–8% CeOx: CeOx content in CeOx/Kaol.

**Table 3 polymers-14-05187-t003:** Processing properties of SBR filled with different contents of CeOx /Kaol material.

CeOx /Kaol (phr)	*M_H_* /(dN·m)	*M_L_* /(dN·m)	ΔM /(dN·m)	*t_10_* /(min)	*t_90_* /(min)
Pure	3.30	0.40	2.90	3.4 ± 0.6	14.5 ± 0.2
10	4.30	0.60	3.70	2.5 ± 0.8	15.0 ± 0.9
20	5.90	0.90	5.00	2.5 ± 0.4	15.3 ± 06
30	8.20	1.30	6.90	5.2 ± 0.8	17.5 ± 0.8
40	13.30	2.30	11.10	5.1 ± 0.6	17.5 ± 0.6
50	8.10	1.20	6.90	5.3 ± 0.9	18.0 ± 0.7
60	9.00	1.50	7.50	5.3 ± 0.7	18.2 ± 0.2
70	8.80	2.40	6.40	5.4 ± 0.4	20.0 ± 0.7
80	9.30	2.60	8.70	6.0 ± 0.6	20.1 ± 0.3

**Table 4 polymers-14-05187-t004:** Static mechanical properties of SBR composites filled with CeOx/Kaol complex loaded different contents of CeOx.

CeOx (%)	Hardness	Tensile Strength/MPa	Tensile Modulus/MPa			Tear Strength/MPa
			100%	300%	500%	
Pure	37 ± 1	2.08 ± 0.07	0.99 ± 0.03	1.69 ± 0.02	-	10.02 ± 0.33
0	56 ± 1	9.45 ± 0.33	1.73 ± 0.08	4.76 ± 0.24	6.64 ± 0.22	36.99 ± 1.59
1	51 ± 1	10.06 ± 0.96	1.03 ± 0.22	2.15 ± 0.67	3.99 ± 0.81	42.51 ± 1.36
2	52 ± 1	9.83 ± 0.99	1.11 ± 0.05	2.48 ± 0.21	4.56 ± 0.31	41.43 ± 3.58
3	52 ± 1	9.50 ± 0.13	1.15 ± 0.05	2.78 ± 0.23	5.18 ± 0.28	42.92 ± 1.19
4	57 ± 1	12.85 ± 0.12	1.33 ± 0.10	3.41 ± 0.31	6.06 ± 0.38	51.16 ± 2.42
5	54 ± 1	11.61 ± 0.14	1.32 ± 0.04	3.26 ± 0.16	5.59 ± 0.25	48.82 ± 1.32
6	53 ± 1	10.49 ± 0.65	1.14 ± 0.06	2.35 ± 0.13	4.05 ± 0.18	41.67 ± 0.62
7	54 ± 1	9.49 ± 0.70	1.10 ± 0.05	2.10 ± 0.16	3.70 ± 0.28	35.63 ± 2.63
8	52 ± 1	9.64 ± 0.25	1.21 ± 0.04	2.34 ± 0.08	4.10 ± 0.16	36.07 ± 2.21

**Table 5 polymers-14-05187-t005:** Static mechanical properties of SBR filled with different contents of CeOx /Kaol material.

CeOx/Kaol (phr)	Hardness/Shore A	Tensile Strength/MPa	Tensile Modulus/MPa			Tear Strength/MPa
			100%	300%	500%	
Pure	37 ± 1	2.08 ± 0.07	0.99 ± 0.03	1.69 ± 0.02	-	10.02 ± 0.33
10	43 ± 1	3.27 ± 0.03	0.73 ± 0.03	1.28 ± 0.10	2.25 ± 0.21	13.74 ± 0.82
20	48 ± 1	5.18 ± 0.50	0.91 ± 0.02	1.87 ± 0.05	3.26 ± 0.09	20.09 ± 1.61
30	54 ± 1	7.47 ± 0.65	1.14 ± 0.01	2.24 ± 0.05	3.84 ± 0.08	23.85 ± 1.83
40	59 ± 1	13.79 ± 0.86	1.88 ± 0.09	4.86 ± 0.28	7.81 ± 0.43	37.45 ± 2.42
50	57 ± 1	12.85 ± 0.12	1.33 ± 0.10	3.41 ± 0.31	6.06 ± 0.38	51.16 ± 1.32
60	58 ± 1	12.16 ± 0.71	1.31 ± 0.03	3.16 ± 0.03	5.69 ± 0.17	50.95 ± 3.36
70	61 ± 1	12.14 ± 0.51	1.58 ± 0.02	4.07 ± 0.09	6.64 ± 0.09	54.20 ± 2.38
80	68 ± 1	12.19 ± 0.55	2.29 ± 0.09	6.85 ± 0.22	10.26 ± 0.21	40.59 ± 2.72

**Table 6 polymers-14-05187-t006:** The anti-ageing properties of the pure SBR and the resulting SBR composites.

Samples	Time/h	Tensile Strength/Mpa (Rate of Change/%)	Shore (A) Hardness (Rate of Change/%)
Pure SBR	0	2.08 ± 0.07	37 ± 1
	24	1.60 ± 0.03 (−18.75%)	41 ± 1 (+ 10.81%)
	48	1.51 ± 0.09 (−27.40%)	45 ± 1 (+ 21.62%)
	72	1.45 ± 0.03 (−30.28%)	47 ± 1 (+27.02%)
Kaol/SBR	0	9.45 ± 0.33	56 ± 1
	24	6.29 ± 0.09 (−33.43%)	62 ± 1 (+5.08%)
	48	3.96 ± 0.19 (−58.09%)	68 ± 1 (+15.25%)
	72	3.79 ± 0.17 (−59.89%)	68 ± 1 (+15.25%)
CeOx/Kaol/SBR	0	13.79 ± 0.86	59 ± 1
	24	7.28 ± 0.23 (−24.48%)	58 ± 1 (+11.53%)
	48	6.08 ± 0.39 (−36.92%)	61 ± 1 (+17.30%)
	72	5.32 ± 0.21 (−44.81%)	66 ± 1 (+26.92%)

## Data Availability

The data presented in this study are available on request from the corresponding author.

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
