# Peer review of "The Improvement of Kaolinite Supported Cerium Oxide for Styrene–Butadiene Rubber Composite: Mechanical, Ageing Properties and Mechanism"

_polymers, 2022, doi:10.3390/polym14235187_

Round 1
Reviewer 1 Report
1- What is the function of styrene-butadiene rubber composite?
2- What is the limiting factor in this study?
3- Add the FTIR spectra for the CeO2 also.
4- The introduction is well-written, but there are some important discussions that are missing in the literature review, the following references can be used:
Mechanical properties and swelling behavior of acrylamide hydrogels using montmorillonite and kaolinite as clays. J. Environ. Treat. Tech, 7, 211-219.
Microneedles for transdermal drug delivery using clay-based composites. Expert Opinion on Drug Delivery, 19(9), 1099-1113.
5- Figure 4. The scale of the SEM images of Ce is not shown in the figure. Is it Ce or CeO2? Why the colors are different? Correct the caption of the figure also.
6- What is a replacement clay for kaolinite? What was the reason to select this clay?
7- It is suggested to add a future prospect at the end of this study.
Author Response
请看附件

Reviewer 2 Report
The work by Liu et al reports on a kaolinite supported cerium oxide for styrene-butadiene rubber. It is an interesting piece of work worth of publication. Nevertheless, I have some suggestions and modifications that should be addressed by authors before its publication.
In experimental section
It is said that “The Kaol sample was wet grounded and the d50 is determined as 1.1μm” How did you determine the d50?
It is said that “The 10% CaCl2 aqueous solution was added into the mixture system using a peri-staltic pump with a rate of 10 ml/min. During the stirring process, the pH value of this mixture system was maintained at 11.0.” How did you maintain constant the pH? Please by more specific.
In Results
In figure 1, authors include the XRD pattern of the pristine kaolinite sample and of the kaolinite sample treated with ceria. In the treated sample, the peaks corresponding to kaolinite are not observed. Does the XRD pattern of kaolinite correspond to original material or to the ground one? In any case, this point should be discussed in the manuscript as mechanical treatments could modify the crystal structure and, therefore, the XRD pattern of the samples. There is some literature about the effect of mechanical treatments on particle size and crystalline structure of clay minerals (1-) that should be discussed here. Thus, both if the mechanical treatment performed by authors does not modify the crystalline structure or if it does modify the structure it is relevant for this work.
1. Franco et al. The influence of ultrasound on the thermal behaviour of a well ordered kaolinite Thermochimica Acta 404 (1-2), 71-79
2. Sanchez-Soto et al. Effects of dry grinding on the structural changes of kaolinite powders. Journal of the American Ceramic Society 83 (7), 1649-1657
3. F. Franco et al, Particle-size reduction of dickite by ultrasound treatments: Effect on the structure, shape and particle-size distribution, Appl. Clay Sci. 35(1-2) (2007) 119-127.
4. J. Arcenegui-Troya et al, Relevance of particle size distribution to kinetic analysis: The case of thermal dehydroxylation of kaolinite, Process. 9(10) (2021).
In the FTIR characterization section, it is claim the crystal degree of Kaol decreased due to loading of CeOx. What do you mean by crystal degree? crystallinity? Please clarify.
The microscopic characterization of the sample is very interesting , but there are too many pictures in the manuscript. I suggest moving some of them to a supplementary section.
In table 2 on processing properties. What is the difference between samples “pure SBR” and “0 CeOx”? please clarify.
Round 2
Reviewer 1 Report
The manuscript is upgraded now and can be published in its present form.
Reviewer 2 Report
Manuscript has been improved and corrected following sugestions. It is ready for publication